# Relationships between structure, in vivo function and long-range axonal target of cortical pyramidal tract neurons

Gerardo Rojas-Piloni[1,2], Jason M. Guest[1,3,4], Robert Egger[4], Andrew S. Johnson[1], Bert Sakmann[1] & Marcel Oberlaender[1,3,4]

Pyramidal tract neurons (PTs) represent the major output cell type of the neocortex. To investigate principles of how the results of cortical processing are broadcasted to different downstream targets thus requires experimental approaches, which provide access to the in vivo electrophysiology of PTs, whose subcortical target regions are identified. On the example of rat barrel cortex (vS1), we illustrate that retrograde tracer injections into multiple subcortical structures allow identifying the long-range axonal targets of individual in vivo recorded PTs. Here we report that soma depth and dendritic path lengths within each cortical layer of vS1, as well as spiking patterns during both periods of ongoing activity and during sensory stimulation, reflect the respective subcortical target regions of PTs. We show that these cellular properties result in a structure–function parameter space that allows predicting a PT's subcortical target region, without the need to inject multiple retrograde tracers.

[1] Digital Neuroanatomy, Max Planck Florida Institute of Neuroscience, 1 Max-Planck-Way, Jupiter, FL 33458, USA. [2] Departamento de Neurobiología del Desarrollo y Neurofisiología, Instituto de Neurobiología, Universidad Nacional Autónoma de México, Boulevard Juriquilla 3001, Campus UNAM-Juriquilla, Querétaro 76230, Mexico. [3] Max Planck Group: In Silico Brain Sciences, Center of Advanced European Studies and Research, Ludwig-Erhard-Allee 2, Bonn 53175, Germany. [4] Bernstein Group: Computational Neuroanatomy, Max Planck Institute for Biological Cybernetics, Spemannstr. 38-44, Tübingen 72076, Germany. Gerardo Rojas-Piloni and Jason M. Guest contributed equally to this work. Correspondence and requests for materials should be addressed to M.O. (email: marcel.oberlaender@caesar.de)

Pyramidal tract neurons (PTs) are canonical elements of the cortical circuitry[1]. Located deep in layer 5 (L5B), they integrate synaptic input patterns that originate from virtually all types of local (intra-cortical (IC)) excitatory and inhibitory neurons, as well as from several cortico-cortical (i.e., across cortical areas) and subcortical (e.g., thalamo-cortical) long-range pathways (reviewed in ref. [2]). Compared to other pyramidal neuron (PN) cell types of the cortex, PTs have sparse IC axon projections[3], suggesting that PTs contribute only little to local computations. In turn, PTs project long-range axons to several subcortical structures[4]. The subcortical targets typically depend on the cortical area the PTs reside in, but vary from cell to cell even within the same cortical area (reviewed in ref. [5]). Pyramidal tract neurons are thus thought to represent one final stage of local cortical circuits, integrating specific combinations of local and long-range input and routing the results of this integration to distinct ensembles of subcortical targets[6].

Layer 5 comprises a second class of excitatory neurons—intratelencephalic PNs (ITs)—which project long-range axons to the striatum and other cortical areas[6–8]. Whereas the morphology and physiology of PTs is broadly consistent within and across cortical areas, it is distinct from that of the neighboring ITs[6]. Furthermore, L5 ITs and PTs have cell-type-specific IC connectivity[9] and brain-wide input patterns[10]. As a result, L5 ITs and PTs were shown to process complementary sensory information, both in mouse primary visual cortex (V1)[10], as well as in rat barrel cortex (vS1)[11, 12]. Recent studies suggested that neurons within the class of ITs may have different functional roles when processing the same sensory stimulus, depending on the cortical area into which they project their respective long-range axons. For example, ITs in L3 of mouse vS1 were shown to project axons either to the vibrissal part of the primary motor (vM1) or secondary somatosensory cortex (vS2). A larger fraction of vS2-projectors (vS2-Ps) than vM1-Ps showed whisker touch-related responses during texture discrimination, whereas the opposite was found during an object detection task[13].

Here we investigated whether long-range target-specific segregation into multiple functional channels—as observed for ITs in L3 of vS1[13] and L5 of V1[14]—also applies to cortical output from the subcortically projecting PTs. The lack of genetic/molecular markers that correlate with the specific long-range targets of PTs[15], as well as the large distances to and between the various subcortical target areas, so far prevented from identifying or reconstructing the long-range targets of PTs after electrophysiological measurements in vivo. To overcome these limitations, we injected different retrograde tracer agents into multiple subcortical areas, performed in vivo cell-attached recordings of the retrogradely labeled PTs and filled them with biocytin to reconstruct their morphologies. We find that soma depth location and layer-specific dendrite distributions allow predicting the respective subcortical target area of PTs, and that spiking patterns during both periods of ongoing activity and during whisker stimulation are target-related. These findings indicate that—similar to ITs—stimulus features may be differentially extracted by PTs via long-range target-specific subnetworks[14], which could be reflected by the target-specific embedding of somata and dendrites into the cortical circuitry.

## Results

### Retrograde injections label different populations of PTs.
First, we determined the respective numbers of PTs that project long-range axons to four of the major subcortical targets of vS1[1, 8, 16, 17] (Fig. 1a): the posterior medial division of the thalamus (POm), the superior colliculus of the tectum (SC), the pontine nucleus (Pons), and the subnucleus caudalis of the spinal trigeminal tract

in the brain stem (Sp5C). We injected Fluoro-Gold (FG-488) into one of the four targets, respectively, and investigated the distributions of retrogradely labeled neurons in consecutive 50 μm thick coronal brain slices. The distance between the center locations of the targets and the injection sites was $231 \pm 121$ μm (median: 186 μm; range: 94–487 μm; $n = 3$ for each target). Injections had a diameter of $2.30 \pm 0.45$ mm, covering $90 \pm 8\%$ of the respective target areas (median: 91%; range: 74–99%; Supplementary Fig. 1). Each subcortical target received input from different cortical areas and layers, but always from L5B of vS1 (Supplementary Fig. 1). Within vS1, we quantified the number of retrogradely labeled PTs for each of the four targets with respect to all neuron somata (i.e., NeuN-647). The resultant soma distributions of PTs overlapped within L5 (Fig. 1b), but peak densities of POm- and SC-projectors (SC-Ps) were ~100 μm more superficial than those of Pons- and Sp5C-Ps (Table 1).

Next, we injected different retrograde tracers (FG-405, Cholera toxin subunit B conjugated to Alexa-594 or Alexa-647 (CTB-594, CTB-647)) into three targets of the same animal ($n \geq 2$ for each possible triple combination). The tracers had very similar labeling efficiencies, as revealed by control experiments in which we simultaneously injected FG and CTB via the same pipette (Fig. 1c). Quantification of the retrogradely labeled neurons in coronal slices of multi-tracer injected brains revealed that, in line with the results from single tracer injections, somata of PTs with different target regions intermingled within L5, but peak densities of the respective soma depth distributions were target-dependent (Fig. 1d). Tangential slices revealed that retrogradely labeled PTs, for any of the injected subcortical target regions, were equally abundant within barrel columns and septa (two-sided $t$-test across 13 slices from 3 rats: $P \geq 0.16$; Fig. 1e). Pons-Ps constitute the largest population of PTs in vS1, followed by Sp5C-, POm-, and SC-Ps (Fig. 1f). A total of $19 \pm 12\%$ of the PTs within each group projected to every other second target (see also ref. [18]), $2 \pm 3\%$ to three targets. Overlap between POm- and Sp5C-Ps was largely absent (see also ref. [19]). Summing the four distributions of retrogradely labeled somata resulted in a vertical density profile (Fig. 1b), which resembled in extent and integral (Fig. 1g) the distribution of L5 PNs with thick-tufted apical dendrites, as reported previously[3, 20].

### Identifying subcortical targets of in vivo recorded PTs.
Because PTs project long-range axons largely to only one of the four major subcortical targets investigated here, the respective number of retrogradely labeled neurons in vS1, when compared to all neurons in L5 is low (Table 1). This is because only ~76% of the neurons in L5 of rat vS1 are excitatory cells[21], and of those, less than 50% are PTs[20]. Furthermore, the labeling efficacy of the retrograde tracers is ~80%[13]. Hence, the low fraction of L5 neurons that can in principle be labeled by injections into a single subcortical target (~5–10%) is likely one of the reasons that so far prevented from in vivo recordings of retrogradely labeled PTs. To overcome this limitation, we combined retrograde injections of FG-405, CTB-594, and CTB-647 into three targets, with in vivo cell-attached recordings. At least 20% of all L5 neurons will be retrogradely labeled in triple injected brains, and blind-patching them becomes more likely. Furthermore, to identify the respective subcortical target region, we labeled each recorded neuron with biocytin[22], which also allowed for post hoc reconstruction of their 3D dendrite morphologies (Fig. 2a). We measured ongoing and sensory-evoked spiking in anesthetized young adult rats ($n = 32$; P33–42), and recovered the morphology of 97 in vivo recorded neurons (see examples in Fig. 2b). Eighty-nine of the labeled neurons were identified as PNs located in L5. In agreement with our estimates above, 48% of the L5 PNs (43 of 89) were

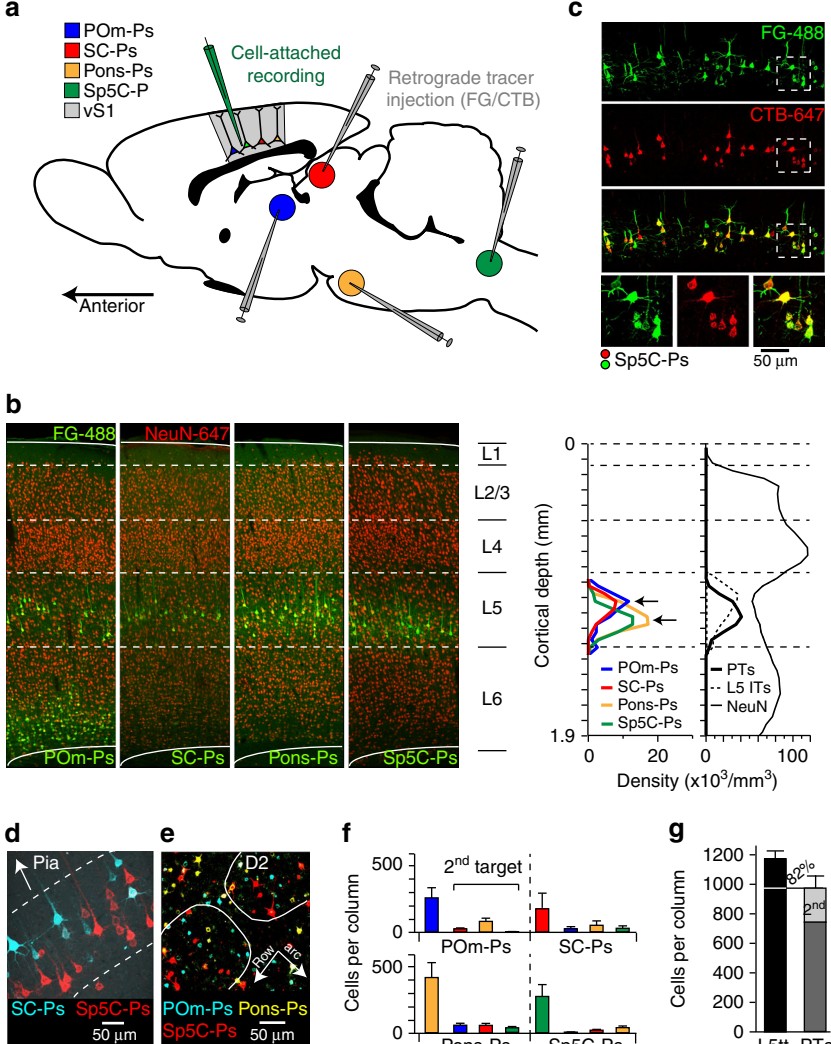

**Fig. 1** PTs innervate different subcortical targets. **a** Injections of retrograde tracers into four of the major subcortical targets of PTs in rat vS1. **b** Left panels: Example images from coronal slices of vS1 for each of the four injected targets. Dashed lines represent layer borders. The respective numbers of PTs (FG-488) were quantified with respect to all neurons (NeuN-647). Right panels: average vertical distributions of PT somata. The vertical profile of ITs represents the distribution of L5 slender-tufted neurons, as determined previously[3, 20]. **c** Example image of vS1 from control experiments in which we injected FG and CTB via the same pipette into the Sp5C. **d** Example image from dual tracer injection illustrates overlapping, target-related sublayers in L5. **e** Example image from tangential slice of vS1 illustrates presence of PTs in barrel columns and septa. **f** Number of PTs per average (C2) barrel column and fractions of PTs projecting to two targets. Error bars denote SDs (applies to all panels and figures). **g** The sum of retrogradely labeled PTs from panel 1f yields that ~82% of the PNs in vS1 with thick-tufted dendrite morphologies project to the four subcortical targets investigated here. Approximately 25% innervate two of the injected targets. The number of L5 thick-tufted PNs per barrel column was determined previously[3, 20]

morphologically identified as PTs (i.e., thick-tufted apical dendrites), of which 47% (20) were labeled by a single retrograde tracer. Three PTs were labeled by two retrograde tracers.

**Dendrite distributions of PTs reflect subcortical targets.** We reconstructed the dendrite morphologies of the in vivo labeled PTs, whose subcortical targets had been identified. The tracing results were augmented with reconstructions of the pial surface, white matter tract, and barrel field in L4. These anatomical structures were used to register each reconstructed morphology to a geometrical reference frame of rat vS1, which allowed determining the 3D soma and dendrite locations with ~50 μm precision (Fig. 3a)[23]. Seven PTs were located in L5A, fifteen in L5B (i.e., above or below 1050 μm; Supplementary Fig. 2). The soma depth distribution across the sample of reconstructed neurons ($n = 22$; range: 913–1244 μm, median: 1090 μm) was similar to

the overall distribution of PTs in vS1 (range: ~900–1300 μm, peak: 1119 μm, Fig. 1b). Soma depth locations were not significantly different between PTs when grouping them by their respective subcortical targets ($n = 5/5/5/4$ for POm/SC/Pons/Sp5C-Ps; one-way ANOVA: $P = 0.76$; Table 1).

For each PT, we extracted a set of 21 morphological parameters (Supplementary Table 1)[20]. Principal component (PC) analysis of the 21-dimensional dendritic feature space (Fig. 3b) revealed that the present sample of PT morphologies ($n = 22$) was indistinguishable from a previously reported sample of in vivo labeled L5 thick-tufted neurons ($n = 16$)[3], but significantly different from the slender-tufted dendritic morphologies of in vivo labeled ITs ($n = 18$; one-way ANOVA of PC1: $P < 0.01$)[3]. Within the population of PTs, the PC that discriminated between thick- and slender-tufted morphologies was neither related to the respective targets ($n = 5/5/5/4$ for POm/SC/Pons/Sp5C-Ps; one-way ANOVA of PC1: $P = 0.12$), nor did it correlate with the PTs'

**Table 1 Target-related structure and in vivo function of PTs**

| Structure (cellular) AVG ± SD | POm-Ps N = 7; n = 3 | SC-Ps N = 5; n = 2 | Pons-Ps N = 7; n = 3 | Sp5C-Ps N = 8; n = 3 | L5 PTs N = 27; n = 3 | L5tt (PTs)[a] | L5st (ITs)[a] |
|---|---|---|---|---|---|---|---|
| Depth (µm) | 1042±105 | 1055±97 | 1131±110 | 1154±98 | 1110±131 | 1154±163 | 1035±165 |
| Cells in L5 (% NeuN) | 7 | 5 | 12 | 9 | 33 | 33 | 43 |
| Cells in L5 (C2 column) | 256±109 | 174±79 | 421±112 | 284±67 | 1135±367 | 1151 | 1500 |
| **Structure (dendritic)** | N = 5 | N = 5 | N = 5 | N = 4 | N = 22 | N = 16[a] | N = 18[a] |
| Total length (mm) | 12.7±1.2 | 13.2±3.9 | 16.8±4.7 | 13.8±3.8 | 14.5±3.7 | 14.7±3.4 | 7.4±1.7 |
| Basal length (mm) | 5.0±0.9 | 5.2±2.0 | 6.2±1.5 | 5.3±2.4 | 5.5±1.7 | 6.8±2.5 | 4.2±1.2 |
| Apical length (mm) | 7.8±0.7 | 8.1±2.3 | 10.6±3.7 | 8.5±1.4 | 9.0±2.5 | 7.9±1.6 | 3.2±0.8 |
| Total branch points (BPs) | 66±3 | 77±18 | 98±24 | 90±15 | 84±19 | 86±23 | 35±11 |
| Basal BPs | 24±5 | 31±11 | 37±6 | 33±14 | 32±10 | 39±15 | 21±8 |
| Apical BPs | 42±3 | 47±9 | 61±19 | 57±4 | 52±13 | 47±12 | 14±6 |
| **Function (ongoing)** AVG ± SE | N = 5 | N = 5 | N = 5 | N = 5 | N = 43 | N = 9[b] | N = 12[b] |
| Spike rate (Hz) | 7.2±1.6 | 1.3±0.6 | 1.8±1.1 | 4.3±1.0 | 3.5±0.5 | 3.5±0.5 | 1.1±0.1 |
| 100 Hz burst (% APs) | 5.2±1.2 | 13.5±6.5 | 5.2±2.0 | 4.6±2.8 | 6.2±1.1 | | |
| 200 Hz burst (% Aps) | 2.1±0.6 | 6.2±3.6 | 0.4±0.3 | 1.5±0.8 | 2.2±0.6 | | |
| **Function (evoked)** | N = 5 | N = 5 | N = 5 | N = 5 | N = 43 | | |
| Onset probability | 0.66±0.09 | 0.23±0.12 | 0.38 ± 0.20 | 0.64±0.11 | 0.45±0.05 | | |
| Onset rate (Hz) | 5.1±1.7 | 1.7±1.2 | 4.5±2.9 | 6.9±3.1 | 4.0±0.8 | | |
| Sustained rate (Hz) | 0.8±1.4 | 2.3±1.1 | 3.3±2.1 | 3.3±1.6 | 2.1±0.8 | | |

[a] and [b] denote previously reported data, refs. [3, 20], respectively. N denotes sample size (i.e., number of brain slices or number of neurons); n denotes number of animals

soma depth ($n = 19$; Pearson: $R = -0.04$). Consequently, the morphology of PTs did not depend on the subcortical target (Supplementary Table 1), or their location within L5A ($n = 7$) or L5B ($n = 15$; two-sided $t$-test: $P = 0.50$). The latter was true for all cellular parameters investigated in this study (Supplementary Fig. 2).

Laminar dendrite distributions (i.e., along the vertical cortex axis) were different across PTs with different target areas (Fig. 3c). To quantify this observation, we calculated the dendritic path length and the number of branch points that each population of PTs contributed to L1, L2/3, L4, L5A, and L5B, respectively. Dendritic path lengths correlated significantly with the respective number of branch points within and across layers (Fig. 3d; $n = 22$; Pearson: $R > 0.84$, $P < 10^{-5}$). Dendrite distributions of POm-Ps were significantly less complex (i.e., quantified as path length times the number of branch points, see also Supplementary Fig. 2) than PTs with different targets ($n = 22$; two-sided $t$-test of dendrite complexity distributions along vertical cortex axis: $P < 0.002$). These differences were primarily due to significantly less dendrites/branch points within L5A ($n = 22$; two-sided $t$-test: $P < 0.007$). In contrast, SC-Ps had more complex dendrite distributions within L5A, compared to PTs with different targets ($n = 22$; two-sided $t$-test: $P < 0.04$), and less complex distributions within L5B ($n = 22$; two-sided $t$-test: $P < 0.001$). Pons-Ps had the most complex dendrite distributions within L1 ($n = 22$; two-sided $t$-test: $P < 0.001$), Sp5C-Ps the least complex distributions in L4 ($n = 22$; two-sided $t$-test: $P < 0.07$).

**Ongoing spike rates of PTs reflect subcortical targets.** Several previous studies have associated PTs with the property of intrinsic bursting (reviewed in ref. [2]). In line with these assessments (e.g., ref. [24]), we found that in our sample, the majority of PTs spiked in bursts of action potentials (APs; 100 Hz: 37/43; 200 Hz: 29/43) during periods of ongoing activity (i.e., without sensory stimulation). The fraction of ongoing spikes that were elicited as bursts was independent of the respective subcortical target ($n = 5$ for each PT group; one-way ANOVA 100 Hz: $P = 0.31$; 200 Hz: $P = 0.18$). Ongoing spike rates in the present

sample of PTs ($n = 43$) were in line with those reported previously[10, 24] and not significantly different for the subset of PTs whose subcortical targets had been identified ($n = 23$; two-sided $t$-test: $P = 0.29$). However, ongoing spike rates reflected the respective subcortical target region (Fig. 3e). POm-Ps were on average significantly more active than PTs with different targets ($n = 20$; two-sided $t$-test: $P < 0.05$). SC- and Pons-Ps had similarly low ongoing spike rates, rendering them as the least active PTs in vS1 ($n = 20$; two-sided $t$-test: $P < 0.004$). Sp5C-Ps had intermediate ongoing spike rates that were lower than those of POm-Ps and higher than those of SC- and Pons-Ps.

**Structure–function relationships predict subcortical targets.** Ongoing spike rates of PTs were shown to be not significantly different during anesthetized and awake conditions[11, 24]. Together with the soma depth location in L5B and the layer-specific dendrite distribution, the present data thus reveals three cellular properties—potentially independent of experimental conditions—that reflect the respective subcortical target region of PTs in rat vS1. We thus calculated the probability distributions for predicting the respective long-range targets based on each of these three properties, respectively (Fig. 4a). The resultant probability distributions were target related for each property, but overlapped, as expected from the variability of the respective target-related properties. The overlaps prevented unambiguous determination of the long-range target for any of the three properties. As a result, classification of the in vivo recorded/reconstructed PTs (see "Methods"), based on their soma depth location or ongoing spike rate or dendrite distribution, failed to identify clusters that were homogeneous with respect to subcortical targets (Fig. 4b).

Next, we plotted the distributions of the three target-related properties in a joint three-dimensional structure–function feature space (Fig. 4c). We found that both soma depth location and dendrite distributions correlate with ongoing spike rates. For example, PTs with ongoing spike rates larger than ~5 Hz that are located within the upper sublayer of L5B (i.e., as formed by somata of PTs with different targets (see Fig. 1)) primarily

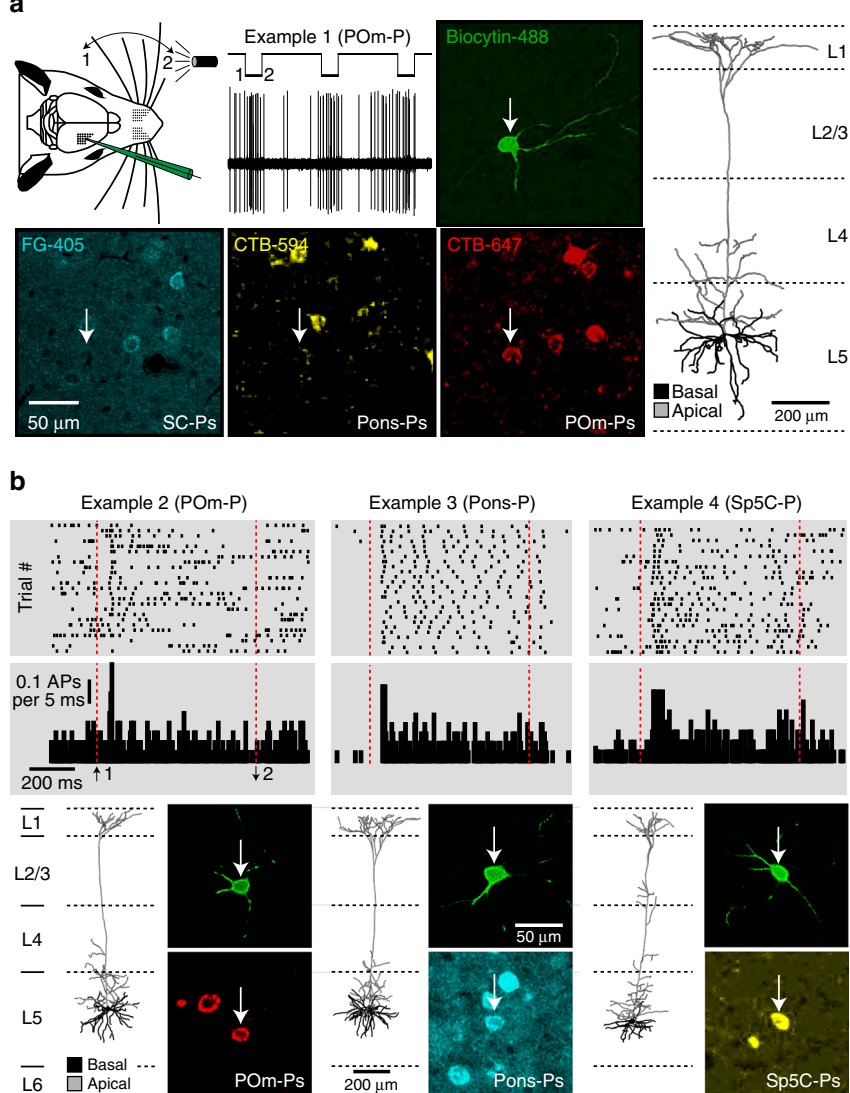

**Fig. 2** Identifying subcortical targets of in vivo recorded PTs. **a** Cell-attached recording and biocytin labeling of exemplary neuron in L5, after retrograde tracer injections into the SC (FG), Pons (CTB-594), and POm (CTB-647). Slices were stained with Alexa-405 and Alexa-488 to reveal FG-positive and biocytin-labeled neurons (white arrow), respectively. Right panel: 3D dendrite reconstruction of the exemplary PT. **b** Top panels: raster plots and PSTHs of spiking activity before (ongoing), during, and after whisker stimulation (red dashed lines) for three example neurons. Bottom panels: 3D dendrite reconstructions and identification of the respective subcortical targets of the three example PTs

represent POm-Ps, whereas those with lower ongoing spike rates project their long-range axon either to the SC or Pons. Analogously in the lower sublayer, Sp5C-Ps could be differentiated by higher ongoing spike rates from SC/Pons-Ps. Consequently, classification based on the combined target-related properties resulted in three clusters, each cluster comprising PTs with largely the same subcortical target (Fig. 4d). Reconstruction of soma depth location and dendrite distribution, when complemented with measurements of the ongoing spike rate, thus allows predicting whether a recorded PT in rat vS1 projects its long-range axon to either the SC/Pons, Pom, or Sp5C, with a confidence level of more than 80% (Fig. 4e).

We repeated our analyses for the three PTs that were labeled with two retrograde tracers. One of the dual-Ps innervated the POm and Pons, two projected to the POm and SC (dual-colored markers in Figs. 3–5 and Supplementary Fig. 2). Each dual-P had structural and functional properties that were related to only one of its respective targets. For example, morphology (Fig. 3d), dendrite distribution (Supplementary Fig. 2), and ongoing spike rate (Fig. 3e) of the POm/Pons-P were consistent with the respective properties of POm-Ps and different from those of Pons-Ps. Consequently, the classification assigned this dual-P to the POm cluster with a confidence of 80% (Fig. 4d). Similarly, the properties of both POm/SC-Ps resembled those of SC-Ps, resulting in an assignment to the SC cluster with a confidence of 100 and 81%, respectively.

**Sensory-evoked spiking of PTs reflects subcortical targets.** Finally, we investigated whether the relationships between soma-dendrite distributions and ongoing spike rate translate into sensory-evoked activity patterns that also reflect a PT's subcortical target region. To do so, we passively deflected all facial whiskers along the caudal axis, by applying a low pressure air puff for 700 ms (see "Methods" section). Stimulus-evoked spiking patterns resembled those reported previously for similar whisker stimuli[25–28] (Fig. 5a). During the first 100 ms of whisker stimulation, spike rates increased significantly, on average to

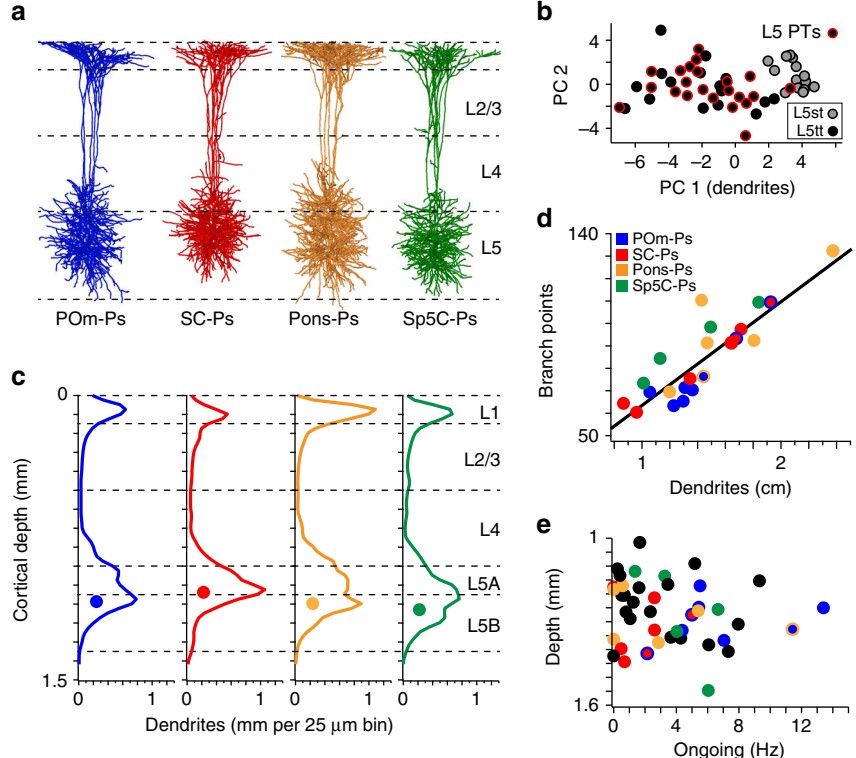

**Fig. 3** Laminar dendrite distributions reflect subcortical targets of PTs. **a** 3D reconstruction of in vivo labeled PTs with identified subcortical target ($n = 5$ for POm/SC/Pons-Ps, $n = 4$ for Sp5C-Ps). PTs were registered to a geometrical reference frame of rat vS1 with ~50 μm precision. **b** Principal components (PCs) of 21 dendritic features (Supplementary Table 1) that discriminate between L5st (ITs) and L5tt PNs (PTs). **c** Distribution of dendrites along the vertical cortex axis, averaged across the PTs shown in **a**. The circles represent the median soma depths of each group. **d** Dendritic path length per PT is significantly correlated to the respective number of branch points. Markers with two colors represent dual-Ps (also applies to Figs. 4 and 5 and Supplementary Fig. 2). **e** Recording depth vs. ongoing spike rate of PTs. Black markers represent PTs with unidentified targets (also applies to Fig. 5)

$7.5 \pm 1.1$ Hz ($n = 43$; two-sided $t$-test, paired: $P < 10^{-5}$). Seventy-nine percent of the PTs in our sample (34/43) responded with increased spiking to the onset of the stimulus. The amplitude of the onset response (i.e., spike rate 0–100 ms post stimulus minus ongoing spike rate) was independent of the long-range target ($n = 5$ for each PT group; one-way ANOVA: $P = 0.48$; Table 1). Response probabilities were also not different between PTs with different targets. Specifically, the fraction of stimulation trials in which PTs increased spiking at short latencies after the onset of the stimulus was similar to those reported previously[27, 28] and did not depend on the subcortical target ($n = 5$ for each PT group; one-way ANOVA: $P = 0.11$). Hence, PTs responded reliably to the onset of the present stimulus, independent of their respective subcortical target.

Following the "onset" responses, spike rates remained significantly elevated ($n = 43$; $5.6 \pm 0.8$ Hz; two-sided $t$-test, paired: $P < 0.03$) for the entire duration of the air puff stimulus, before returning back to baseline ~100 ms after the stimulus had ended. These "sustained" responses were related to the respective subcortical target. To quantify these differences, we calculated two indices (SIs) that allow quantifying the similarity (see "Methods" section) between the PSTH of each individual PT and the four PSTHs average across PTs with the same target (Fig. 5b). The similarity analysis revealed that sustained spiking responses were more similar ($n = 5$ for each PT group; one-way ANOVA of SI1/2: $P < 0.0001/0.0004$) when PTs had the same target (Fig. 5c). In POm-Ps, spike rates during the sustained responses were comparable to those during ongoing periods, which was significantly different compared to PTs that did not project to the POm ($n = 20$; two-sided $t$-test of SI1 + 2: $P < 0.002$). In

contrast, Sp5C-, SC-, and Pons-Ps showed increased activity throughout the sustained responses ($n = 5$ for each PT group; Tukey HSD of SI1: $P < 0.01/0.05/0.1$ for POm-Ps vs. Sp5C/SC/Pons-Ps). Specific for Sp5C-Ps, sustained spike rates increased after about 500 ms, whereas spike rates remained largely constant in SC-Ps throughout the stimulation and decreased with stimulus duration in Pons-Ps (Tukey HSD of SI1: $P < 0.01/0.01$ for Sp5C-Ps ($n = 5$) vs. SC/Pons-Ps ($n = 10$)). Sustained spiking patterns of SC- and Pons-Ps were more similar to each other, when compared to POm- and Sp5C-Ps (Fig. 5d), which was also true for their ongoing spike rates (see classification above). Nonetheless, sustained spiking patterns of SC- and Pons-Ps differed significantly from each other ($n = 10$; Tukey HSD of SI2: $P < 0.01$). Dual-Ps had evoked spiking patterns that were most similar to those PTs, whose cluster they were assigned to (Fig. 5c, d). Combining the stimulus-independent feature space with responses to the present stimulus increased the confidence level of the clustering to 95% (including dual-Ps; but see Supplementary Data 1).

## Discussion

We present an experimental approach to measure the structure and in vivo electrophysiology of long-range projection neurons with respect to their target areas. The data reveals that laminar soma-dendrite distributions and spiking activity of individual PTs are correlated to the subcortical target areas of their respective long-range axons. This does however not imply that PTs innervate only a single subcortical area. A recent study in mouse vS1 showed that ~40% of the PTs project to two subcortical areas, ~20% to three targets[18]. The additional targets can comprise

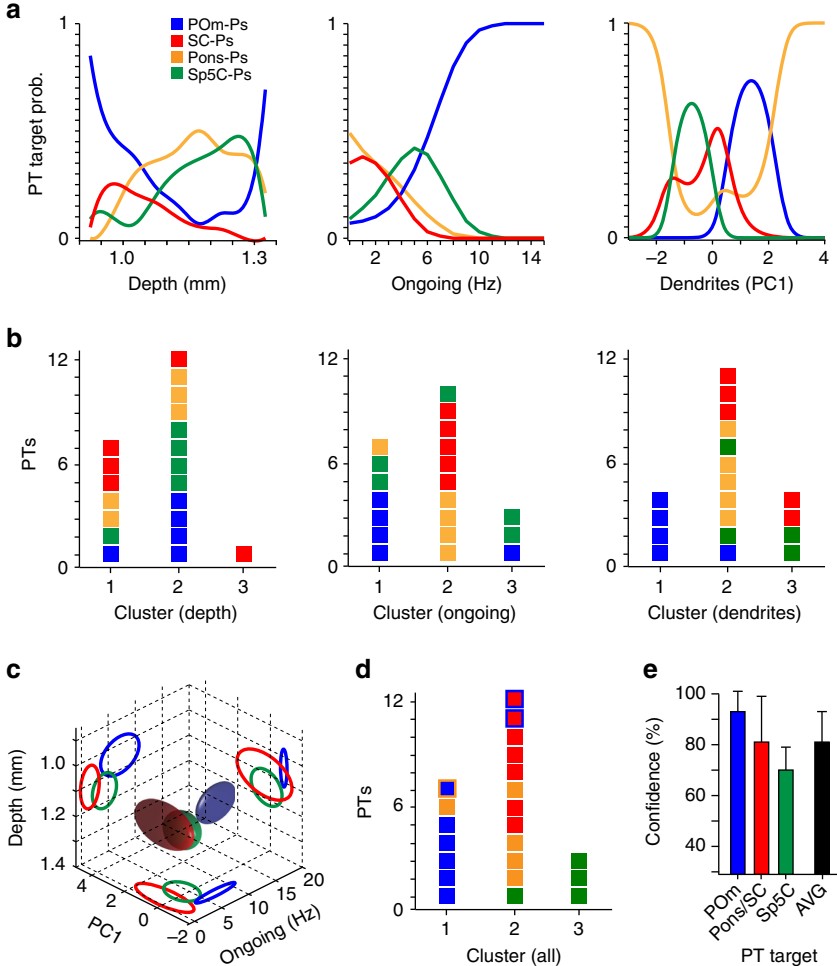

**Fig. 4** Structure–function relationships predict subcortical targets of PTs. **a** Probabilities that PTs project to each of the four subcortical targets, depending on their soma depth location, ongoing spike rate, or dendritic properties (from left to right). **b** Classification of PTs by each of the target-related properties yield clusters that are heterogeneous with respect to subcortical targets. **c** Combining the three target-related properties from panel a yields a 3D parameter space, where PTs with different targets form largely non-overlapping ellipsoids that represent the means ± SDs across PTs with the same target (blue: POm-Ps, green: Sp5C-Ps; red: Pons- and SC-Ps). **d** Classification of PTs using all three target-related properties (dual-Ps are outlined). **e** Confidence of predicting the subcortical targets of PTs

different subnuclei within the four main target regions of vS1 (i.e., thalamus, midbrain, pons, and spinal trigeminal tract). For example, axon terminals of PTs in vS1 cluster within the principal and caudal interpolaris nuclei of the spinal trigeminal tract, in addition to those innervating the Sp5C[29]. Other additional targets comprise a variety of subcortical regions throughout the brain, such as the hypothalamus[18]. Consequently, because PTs that project their long-range axons into the four subcortical targets investigated is this study represent more than 80% all PTs in vS1 (see also ref. [18]), it is to be expected that the majority of them innervates additional targets. However, PTs that for example project to the POm thalamus, do typically not have projections to the SC, Pons, or Sp5C, and vice versa (see also ref. [19]; but see Supplementary Table 2 for the number of PTs per barrel column that project to two of the injected targets).

One possible explanation for the target-related spiking patterns could be that PTs are subdivided into multiple subtypes, which depending on the targets, differ in their genetic/molecular identity, morphology, intrinsic physiological properties, and/or combinations thereof. However, previous studies, as well as the present data, support the notion that PTs represent a single class of cells. First, genetic approaches failed to reveal long-range projections to specific subcortical targets within the group of PTs[15]. For example, neurons that express glt or thy-1, typical markers that differentiate PTs from ITs in L5, were shown to innervate all of the four subcortical target areas investigated here[18, 19, 30]. Second, we show that dendrite morphologies of PTs are not related to the subcortical targets, but differ from those of L5 ITs. Third, somatic current injections revealed significant differences of intrinsic properties between L5 ITs and PTs[19]. Within the class of PTs, intrinsic properties were however largely independent of their respective subcortical target[19]. Supporting these observations, we find that the majority of PTs generate bursts of APs during periods of ongoing activity, independent of the respective subcortical target area.

Even though the intrinsic properties of PTs are largely target-independent, there is one exception. POm-Ps display stronger potassium-mediated hyperpolarizing currents, compared to PTs with different targets[19]. Our data indicate that dendritic path lengths and numbers of branch points of POm-Ps were in general lower than those of PTs with different targets. Thus, within the class of PTs, POm-Ps are not only most specific in terms of their intrinsic physiological properties, but also with respect to their dendritic morphologies. In combination, these properties may in part underlie the observation that POm-Ps have in vivo spiking patterns that are most distinct from those of other PTs. They

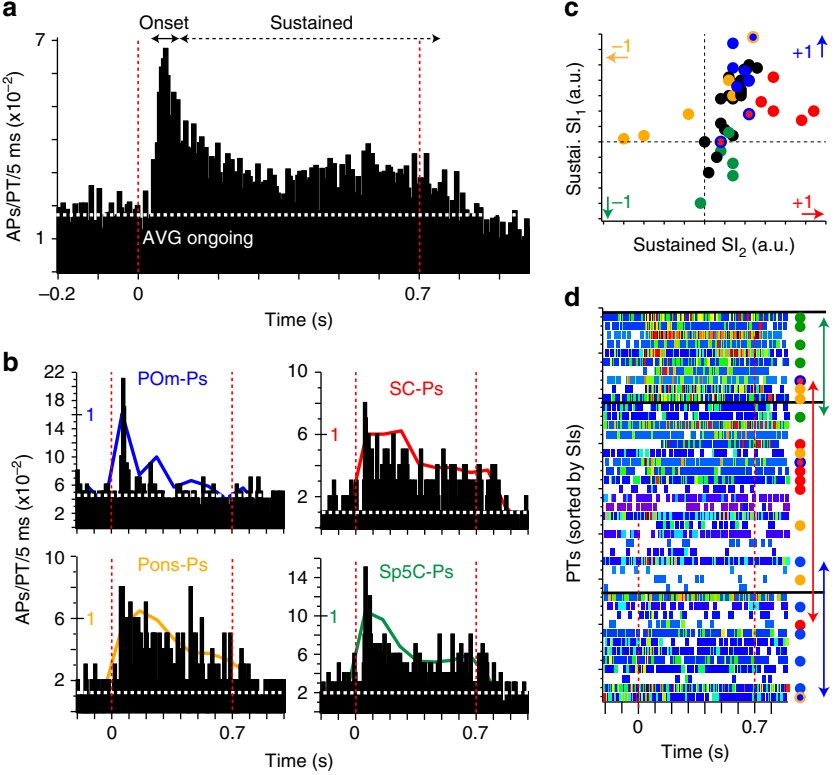

**Fig. 5** Sensory-evoked spiking reflects subcortical targets of PTs. **a** PSTH averaged across 43 PTs for passive multi-whisker deflections. **b** PSTHs averaged across PTs with the same target. Colored lines: PSTHs with 100 ms resolution and normalized to peak response. **c** Similarity indices (SIs) between PSTHs of each individual PT. **d** PSTHs (in 5 ms bins) of each PT sorted by their SIs (color map maximum (red): ≥50 Hz). Colored markers (right) represent those PTs whose subcortical targets were identified. Horizontal black lines denote approximate separation between the populations of Sp5C-, SC/Pons-, and POm-Ps (i.e., colored arrows denote the true, overlapping ranges)

display the highest ongoing spike rates and lack sustained spiking patterns after multi-whisker air puff stimulation.

We conclude that PTs projecting to different subcortical targets do not represent different cell types. Nonetheless, several cellular properties of PTs, which go beyond those that discriminate them from L5 ITs, were related to the long-range target: PTs form two overlapping sublayers in L5B, display significantly different ongoing spike rates and have layer-specific dendrite distributions. Classification based on these parameters revealed that the three target-related properties correlate with each other—potentially reflecting a common origin. Specifically, the differences in soma depth location, in combination with the target-related laminar distribution of dendrites, are likely to reflect differences in synaptic input that PTs receive. Layer-specific axon innervation is a hallmark of cortical organization, observed consistently for local populations (e.g., ref. [3]), as well as for long-range axons (e.g., ref. [31]). The respective extent to which the dendrites of PTs overlap with these layer-specific axon distributions will hence correlate with their long-range target. As a result, target-related differences in the populations that provide input to PTs and/or in the dendritic locations of synaptic input patterns from the same input populations, which may be further amplified by specific biophysical properties of dendritic subdomains[32], are expected to yield different spiking patterns. Our finding that the relationship between ongoing spike rate and long-range axonal targets extends to sensory-evoked spiking patterns—even for simple whisker stimuli, such as passive deflections—supports the conclusion that PTs may be embedded into the cortical circuitry in a target-related manner.

We speculate that a developmental mechanism could result in target-related properties of PTs. Previous studies showed that L5

PNs select the targets of their axons some days before birth, a process that is finalized as early as postnatal day 3[33]. However, L5 PNs were shown to initially develop the same basic set of collateral branches, independent of whether the projection formed is functionally appropriate[34]. Therefore, PTs may start out by innervating all (or multiple) of the subcortical targets investigated here. Axonal innervation of multiple target structures, which is then followed by activity-dependent pruning is one of the key mechanisms during development (reviewed in ref. [35]). Therefore, axons of PTs may be pruned, and only those connections remain that extract information from sensory stimuli that is most appropriate and/or relevant for the respective targets. For example, projections from vS1 to the Sp5C have been suggested to be involved in whisker motor control[36], whereas those to the POm (i.e., higher-order thalamus) are often referred to as driver inputs[37, 38] to transthalamic pathways (i.e., cortico-thalamo-cortical[39, 40]). Because the transthalamic branch often also innervates subcortical motor centers, POm-Ps have been suggested to give rise to efference copies[39]. The process of "function-specific" pruning may not be exclusive to PTs, but could also underlie target-related responses within the groups of ITs in L3[13] and L5[14], and may extend from the targets investigated here, to specific combinations of multiple target areas[18].

Our experiments were carried out during the fifth and sixth postnatal week, at time points when the development of PTs is close to maturation[2]. The low number of PTs projecting to two of the investigated targets (see also refs. [18, 19]), and the even lower number of neurons with three targets, may hence reflect the last stages of this pruning process—a hypothesis that is in line with the properties of the dual-Ps reported here. Future studies need to investigate whether the target-related responses indeed represent

different features of the same sensory stimulus, and whether the extraction of specific stimulus features via target-related soma-dendrite distributions results in stabilizing connections to one (or multiple) of the subcortical targets, and pruning of long-range axons otherwise. Our approach provides a general strategy for such investigations, and the present classification results will help to facilitate functional studies in rat vS1, allowing to discriminate between SC/Pons-, POm-, and Sp5C-Ps without the need to inject retrograde tracers (the routines to predict these targets can be obtained from Supplementary Data 1).

## Methods

**Retrograde tracer injections.** All experimental procedures were carried out after evaluation by a local ethical committee (IACUC) and in accordance with the animal welfare guidelines of the Max Planck Society. Animals were housed in a vivarium with normal day/night cycle and in groups of two–four animals per cage. Young adult (P28–P35) male Wistar rats were injected with 1 mg/ml buprenor-phine SR (0.05cc, s.q.) approximately 30 min prior to surgery, then anesthetized with a ketamine–xylazine mixture (70/6 mg/kg, i.p.) and supplemented with iso-fluorane/O₂ gas. Rats were then placed in a stereotaxic frame (Kopf Instruments, model 1900) and given an injection of .25% bupivacaine (0.10cc, s.q.) at the incision site. Then a 5 cm incision across the midline, just past the base of the neck was made to expose the skull. Both bregma and lambda were located and marked with a surgical pen. A small craniotomy (or craniotomies for multiple site injection experiments) in the skull was made using a dental drill (Osada, model EXL-M40) over the injection site of the right cerebral hemisphere (for Sp5C injections no craniotomy was necessary). Injection site coordinates were as follows (in mm): POm: 2.1 lateral from midline, 3.25 posterior to bregma and 5.2 deep from the pia; SC: 1.3 from the midline, 2.0 anterior to lambda and 3.5 deep from the pia; Pons: 1.1 from the midline, 1.8 anterior to lambda and 8.5 deep from the pia. Prior to injecting tracers into the Pons, SC, and POm, the head of the rat was leveled with a precision of 1 μm in both the medial–lateral and anterior–posterior planes using an electronic leveling device (eLeVeLeR; Sigmann Elektronics, Hüffenhardt, Germany) mounted to an adapter for the Kopf stereotax. The distance of the two aluminum probes of the leveling piece was adjusted to the distance of bregma and lambda. The probes were lowered onto the skull and placed on the markings for bregma and lambda. The tilt of the rats head was then adjusted until both sensors showed the same relative distance. The leveler was raised so the probes were no longer touching the skull, the leveler was turned 90° and the probes were lowered back to the surface of the skull. The same procedure was then repeated to adjust the coronal tilt. Injections into Sp5C of the left side of the brain stem were performed through the atlanto-occipital foramen, 3.0 mm from the midline, 1.4 mm depth at the obex level. Retrograde tracers were pressure injected (50–200 nL) under visual control using a 30cc syringe coupled to a calibrated glass injection capillary. After injection of tracers, the incision site was thoroughly cleaned with saline and sutured. Injections into multiple targets of the same animal were performed using combinations of three retrograde tracers: FG (Fluorochrome; 3% in distilled water), CTB-594 and CTB-647 (Molecular Probes; 1 mg/ml in PBS). In 30 (FG: 3/CTB-594: 19/CTB-647: 8), 24 (23/1/0), 17 (4/7/6), and 23 (1/4/17) of the triple injected animals (i.e., those used for in vivo recording experiments), injections were targeted at the POm, SC, Pons, or Sp5C, respectively. Rats underwent a 5–7-day incubation period after tracer injection before transcardial perfusion or in vivo electro-physiology experiments. The locations of the injection sites and their respective overlap with the target regions were quantified for those experiments used to determine the number and distribution of PTs as shown in Fig. 1 (i.e., single tracer FG injections, n = 3 for each target). To do so, low-resolution images of the respective injection sites were aligned and scaled to match the corresponding images from the Paxinos atlas[41] (number of coronal figure: POm: 36, SC: 43, Pons: 52, Sp5C: 77). Tracing of the target areas and injection sites, as well as calculating the respective overlap between them was performed using the FilamentEditor[42] in Amira Software[43]. Contours representing the overlaps were averaged for each target using the FilamentEditor. The resultant "average" injection sites and center locations are shown with respect to the target area contours from the atlas in Supplementary Fig. 1b.

**In vivo recordings.** In vivo cell-attached recordings and biocytin fillings have been described in detail previously[22, 44]. Briefly, (retrograde tracer injected) rats were (re-)anesthetized with isoflurane and subsequently with urethane (1.4 g/kg) by intraperitoneal injection. The depth of anesthesia was assessed by monitoring pinch withdrawal, eyelid reflexes, and vibrissae movements. Throughout the experiment, the animal's body temperature was maintained at 37.5 ± 0.5 °C by a heating pad. A cranial window with a size of 2 × 2 mm was made 2.1 mm posterior and 5.5 mm lateral to the bregma on the right cerebral hemisphere above vS1. Patch pipettes were prepared from borosilicate glass with a pipette tip diameter of 1 μm (5–8 MΩ) and were filled with normal rat ringer supplemented with 2% biocytin (Sigma: 576-19-2). The pipette was advanced in 1 μm steps to locate single neurons, which was indicated by an increase in electrode resistance (unbiased sampling, irrespective of spiking activity). At this stage, AP waveforms were

recorded using an Axoclamp 2-B amplifier (Axon Instruments, Foster City, CA) or an extracellular loose patch amplifier (ELC-01X, npi electronic GmbH) and digi-tized using a CED *power*1401 data acquisition board (CED, Cambridge Electronic Design, Cambridge, UK). Subsequently, the pipette was advanced until the resis-tance was 25–35 MΩ and APs had an amplitude of 3–8 mV. Ongoing and sensory-evoked spiking of each neuron was recorded during 20–30 trials of passive multi-whisker deflections. Specifically, a plastic tube (1 mm tip diameter) was placed at a distance of 8–10 cm from the whisker pad and delivered an air puff (10 PSI), which deflected the principal and all surrounding whiskers along the caudal axis for 700 ms. Stimulation was repeated at constant intervals (0.3 Hz) and occurred randomly with respect to up- and down-states. Following the recording, juxtasomal biocytin filling was performed by applying continuous, low intensity square pulses of positive current (<7 nA, 200 ms on/200 ms off), while gradually increasing the current in steps of 0.1 nA and monitoring the AP waveform and frequency. The membrane opening was indicated by a sudden increase in AP frequency. Filling sessions were repeated several times (10–40 min) and diffusion was allowed for 1–2 h to obtain high-quality fillings.

**Histology.** Rats were perfused transcardially with phosphate buffer, and brains were removed and fixed with paraformaldehyde. For counting of FG-positive somata in single tracer injection experiments, consecutive 50 μm thick vibratome slices were cut coronally through vS1. Slices were double-immunolabeled to count retrogradely labeled cells (FG) with respect to all neurons (NeuN[45]). To do so, slices were permeabilized and blocked in 0.5% Triton X-100 (TX) (Sigma Aldrich #9002-93-1) in 100 mM phosphate buffer (PB) containing 4% normal goat serum (NGS) (Jackson ImmunoResearch Laboratories #005-000-121) for 2 h at room temperature. Primary antibodies were diluted 1:500 (mouse anti-NeuN, EMD Millipore #MAB377) and 1:500 (Rabbit anti-FG, EMD Millipore #AB153) in PB containing 1% NGS for 48 h at 4 °C. Secondary antibodies (1:500 goat anti-mouse IgG1 Alexa-647 and 1:500 goat anti-Rabbit Alexa-488 molecular probes #A-21240, #A11008) were incubated for 2–3 h at room temperature in PB containing 3% NGS and 0.3% TX. For counting of retrogradely labeled somata in triple tracer injection experiments, cortex was cut either coronally or tangentially to vS1 (45°) into 50 μm thick consecutive slices. Slices were labeled with the above procedure, excluding anti-NeuN/Alexa-647. Further, goat anti-rabbit Alexa-488 was replaced with goat anti-rabbit Alexa-405 (Molecular Probes #A-31556) to stain FG-positive neurons. In experiments where triple retrograde injections were combined with in vivo recording and biocytin filling, cortex was cut into 45–48 consecutive 50 μm thick tangential slices, which were labeled with the above procedure (FG-405), but were additionally treated with Streptavidin Alexa-488 conjugate (5 mg/ml Molecular Probes #S11223) in PB with 0.3% TX for 3–4 h at room temperature to stain biocytin-labeled morphologies. All slices were mounted on glass slides, embedded with SlowFade Gold (Invitrogen) and enclosed with a cover slip.

**Image acquisition.** All images were acquired using a confocal laser scanning system (Leica Application Suite Advanced Fluorescence SP5; Leica Microsystems) equipped with glycerol/oil immersion objectives (HC PL APO 10× .04N.A., HC PL APO 20× .7N.A., and HCX PL APO 63 × 1.3N.A.), a tandem scanning system (Resonance Scanner: 8 kHz scanning speed), spectral detectors with hybrid tech-nology (GaAsP photocathode; 8× line average) and mosaic scanning software (Matrix Screener, beta version provided by Frank Sieckmann, Leica Microsystems). For imaging of the four respective Alexa-Fluors the following settings were used: Alexa-405 (excitation: 405 nm (UV-laser); emission detection range: 400–455 nm), Alexa-488 (488 nm (Ar-laser); 495–550 nm), Alexa-594 (561 nm (DPSS-laser); 600–630 nm), Alexa-647 (633 nm (HeNe-laser); 650–740 nm). For single tracer injection experiments (FG-488, NeuN-647), dual channel mosaic scans of areas up to 7.5 × 10 mm were acquired using a ×10 glycerol objective at a resolution of 0.868 × 0.868 μm per pixel (×1.7 digital zoom, ~20 × 30 fields of view). For triple tracer injection experiments (FG-405, CTB-594, CTB-647), sequential channel mosaic scans of areas of up to 7 × 12 mm were acquired using a ×20 glycerol objective at resolution of 0.361 × 0.361 μm per pixel (×2.0 digital zoom, ~22 × 36 fields of view). For experiments where triple tracer injections were combined with biocytin labeling (biocytin-488, FG-405, CTB-594, CTB-647), sequential channel images of single fields of view were acquired using a ×20 glycerol objective at resolution of 0.361 × 0.361 μm per pixel (×2.0 digital zoom, 1 × 1 fields of view). For 3D reconstruction of dendrite morphologies (biocytin-488), single channel mosaic image stacks of volumes up to 0.8 × 0.8 × 0.05 mm were acquired using a ×63 glycerol objective at a resolution of 0.092 × 0.092 × 0.5 μm per voxel (×2.5 digital zoom, 10 × 10 fields of view). Image stacks were acquired for ~30 consecutive 50 μm thick brain slices to cover complete dendrite morphologies from the pia to L6.

**Detection of somata.** Retrogradely labeled neuron somata were detected manually in high-resolution confocal images using Amira Software[43]. In triple tracer injected animals (i.e., 2× CTB, 1× FG; n ≥ 2 per possible triple combination), retrogradely labeled somata were marked in each image channel (i.e., Alexa-405, Alexa-594, Alexa-647) separately. Double- and triple-labeled neurons were determined as those that were marked in two and three channels, respectively. NeuN-labeled somata were detected automatically using previously reported custom-designed

software[46]. Images from coronal slices were aligned such that the vertical cortical axes were parallel before marking soma locations. Soma distributions from each slice were then converted into 1D density profiles along the vertical cortical axis by summing all somata in 50 μm intervals. One-dimensional profiles were aligned vertically using the L4 peak of the NeuN profiles in each respective slice, and the aligned 1D NeuN, as well as FG profiles were averaged for each of the four targets. To estimate the total number of retrogradely somata per average (C2) barrel column, the present NeuN profile was aligned and uniformly scaled to match the 1D profile of the 3D density of NeuN-positive somata determined previously[21]. The scaling factor was determined by minimizing the squared error between the respective NeuN profiles. The 1D profiles of FG-positive somata were scaled accordingly.

**Reconstruction of dendrites**. Three-dimensional reconstruction of dendrite morphologies was based on a previously described method for semi-automated reconstruction of neuron morphology from brightfield microscope images[47]. Here, the tracing software was adapted to confocal image data, acquired at a lateral/axial resolution of 0.092/0.5 μm as described above. Image stacks were deconvolved using a linear Tikhonov–Miller algorithm and a theoretically computed point spread function of the confocal microscope[48] using Huygens software (SVI, the Netherlands). Neuronal structures were automatically detected in the image stacks using custom-designed software[47]. Proof editing of the automated tracing results, as well as alignment and splicing of neuronal branches across consecutive histological slices was done by using the FilamentEditor[42] (without knowledge of the PTs' subcortical targets). Three-dimensional dendrite reconstructions were augmented with contours of anatomical reference structures (pial surface, white matter tract and L4 barrels), which were drawn manually in low-resolution (×4) images of all consecutive histological slices.

**Physiology analysis**. Ongoing and stimulus-evoked spiking was recorded with Spike-2 software (CED, Cambridge) or with custom-written LabVIEW routines (National Instruments Limited). Spiking profiles during periods of ongoing activity and during air puff stimulation were calculated offline by Spike-2 software or with custom-written MATLAB routines (Math Works, USA). To determine ongoing activity, spontaneously occurring spikes were detected during 200 ms before stimulating the whiskers. Ongoing and sensory-evoked spikes were measured for 20–30 whisker deflection trials and the resultant PSTHs were analyzed at a temporal resolution of 5 ms bins. Similarity between sensory-evoked responses was calculated as follows: (1) Ongoing spike rates were subtracted from each 5 ms bin of the four target-related average PSTHs and PSTHs of each individual PT. If subtraction of ongoing spiking resulted in negative values, the respective bins were set to zero; (2) Each resultant PSTH of sensory-evoked spiking was then normalized to the respective bin with maximal spike rates; (3) For each PT, the bin-wise absolute difference between its normalized sensory-evoked PSTH and each of the four target-related PSTHs was calculated; (4) The bin-wise differences were summed across the entire recording period (−200 to 1100 ms post stimulus). This sum was defined as the similarity between PSTHs (i.e., the smaller the similarity value, the more similar the PSTHs). The four similarity values of each PT were combined to two similarity indices: as shown in Fig. 5c: $\frac{\text{Similarity}_{\text{to Pons}} - \text{Similarity}_{\text{to SC}}}{\text{Similarity}_{\text{to SC}} + \text{Similarity}_{\text{to Pons}}}$ (x-axis); $\frac{\text{Similarity}_{\text{to Sp5C}} - \text{Similarity}_{\text{to POm}}}{\text{Similarity}_{\text{to Pom}} + \text{Similarity}_{\text{to SP5C}}}$ (y-axis).

**Assignment to morphological cell types**. Using the reconstructions of anatomical reference structures (see above), all reconstructed dendrite morphologies of PTs were registered to a standardized 3D reference frame of rat vS1[23]. The shortest distance from the pia surface to the soma location (i.e., cortical depth), as well as 20 morphological and topological features that have previously been shown to separate between morphological types of excitatory PNs in L5/6 of rat vS1[20] were calculated for each reconstructed and registered dendrite morphology. The 21 parameters of each PT were compared to the same parameters obtained from L5tt and L5st dendrite morphologies of in vivo labeled PNs, reported previously[3]. A principal component analysis of the 21-dimensional feature space was performed, and the first two principal components (PC) were plotted in Fig. 3b. Biocytin-filled neurons that were not retrogradely labeled were assigned to L5 ITs or PTs based on dendrite morphology, by comparison with previously reported morphologies of the respective cell types[3, 20].

**Clustering of PTs**. Identified PTs were assigned to three clusters based on their subcortical projection target (i.e., POm-Ps, Sp5C-Ps, and Pons-/SC-Ps). To assess if these clusters are well separated based on different functional and structural parameters, we constructed a three-dimensional probability space that allows determining the cluster which a single neuron is most likely to be part of (see also ref. [19]). Specifically, we calculated the mean and variability (i.e., the covariance matrix) of the parameters for each cluster, and then measured the distance of each individual PT to the center of each cluster, in units of the standard deviation of the cluster. Each individual neuron was then assigned to the cluster with the minimum

distance, and the confidence of assignment to cluster $k$ was calculated as:

$$P(k) = \frac{p_k}{\sum_{\text{clusters } i} p_i}, \text{ where } p_i = 1 - F\left(d_i^2, \text{DOF}_i\right)$$

is the probability of finding a single neuron at distance $d_i$ from cluster $i$, given the degrees of freedom $\text{DOF}_i$ of that cluster (i.e., the number of parameters), and where $F$ represents the cumulative distribution function of the chi-squared distribution. This procedure was repeated independently using only the cortical depth of the soma, ongoing spiking, or features of the dendrite morphology as parameters, and by combining the three parameters. Finally, we repeated the classification by adding the PSTH SIs to the stimulus-independent feature space.

**Code availability**. The target prediction tool can be downloaded from the Supplementary Material (Supplementary Data 1). Instructions how to install and use the tool can be obtained from a README.txt file that is provided with the tool.

**Data availability**. All relevant data are available from the authors.

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

## Acknowledgements

We thank Simon Broghammer for reconstructing neuron morphologies, Christopher Tull for adapting tracing algorithms to confocal images and Damian J. Wallace for comments and discussions on the manuscript. We also thank Alexander Borst, David Fitzpatrick, and Christiaan P.J. de Kock for comments on previous versions of the manuscript. Funding was provided by the Fulbright Scholar Program and CONACyT Fronteras de la Ciencia 2015-02-846 (to G.R.-P.), the Max Planck Institute for Biological Cybernetics (M.O.), the Center of Advanced European Studies and Research (M.O.), the Studienstiftung des deutschen Volkes (R.E.), the Bernstein Center for Computational Neuroscience, funded by German Federal Ministry of Education and Research Grant BMBF/FKZ 01GQ1002 (to M.O.), the European Research Council (ERC) under the European Union's Horizon 2020 research and innovation program (Grant agreement No 633428) (to M.O.), and the Max Planck Florida Institute for Neuroscience (G.R.P., J.M.G., and A.S.J.).

## Author contributions

M.O. conceived and designed the study. G.R.-P. and J.M.G. carried out experiments with advice from B.S., R.E. and A.S.J. carried out morphological classifications. All authors performed data analysis. M.O. wrote the paper.

## Additional information

**Competing interests:** The authors declare no competing financial interests.

