## [Peer Review File · Nature Communications]

Reviewers' comments:

Reviewer #1 (Remarks to the Author):

In this manuscript, Rojas-Piloni, Guest et al. investigate how the structure and activity of pyramidal tract (PT) neurons in layer 5 (L5) of rat barrel cortex relate to their long-range projection targets. This is an important problem within the larger context of understanding how brain structure and circuitry relate to function. The authors injected retrograde tracers---up to 3 in individual animals----into three major subcortical target regions. In addition, they performed in vivo cell-attached electrophysiological recordings and filled recorded neurons with biocytin to reconstruct their soma location and dendritic morphologies with respect to the cortical layers, and to relate associated features to the long-range projection targets of each neuron.

The authors find that PT neurons that project to the superior colliculus (SC-Ps), to the Pons (Pons-Ps), posterior-medial thalamus (Pom-Ps), and spinal trigeminal nucleus caudalis (SP5C-Ps) are largely overlapping in most individual properties, but contain distinct groups when their properties are considered together (i.e. when clustered in multidimensional space).

First, the four types of PTs differ in the distributions of their soma locations within L5. For instance, SC-Ps tend to have a more shallow soma location, although the distributions were highly overlapping.

The dendritic morphology also differed among classes. The authors first define morphology in terms of a first principal component (PC1) from principal component analysis (PCA) on a set of 21 metrics related to dendrite structure. This PCA distinguished the PTs of the present study from thin-tufted intratelencephalic (IT) L5 neurons, but did not distinguish among the subclasses of PT. However, a single measure---the distribution of dendritic length across cortical depth----did distinguish. (Dendritic length and the number of branch points were highly correlated, and offer a measure of dendrite complexity.)

The spontaneous spike rates differed by PT type, with Pom-Ps showing higher levels of activity. Airpuff stimulus-evoked activity also differed among types, with all types showing similar stimulus onset responses, but different patterns of responses to the sustained and offset portions of the stimulus.

The authors show that no one individual feature of the morphology or activity allows classification of the PTs by projection target, but that clustering in three dimensions---based on soma depth, PC1, and spontaneous activity---allow PT types to be classified with an accuracy of about 80%. This means that other investigators can with reasonable confidence infer the projection target of recorded PTs after recording spontaneous activity and reconstructing local (within column) morphology, without having to perform difficult retrograde labeling procedures such as those employed here.

The authors offer Python-based analysis code to explicitly document, and to help readers apply, the classification method. I did not evaluate the code and associated files in detail, but the code looks clear and well documented, and its inclusion as part of this manuscript is a notable strength.

The manuscript addresses an important problem. The quality of the data and analyses appear high. The manuscript is clearly written. I am happy to report that I have no major concerns, and that I think this will make a strong contribution.

I have only two minor suggestion:

1. Clustering based on properties that do not require delivery of stimuli---i.e. spontaneous spike rate,

depth, and morphology---makes sense, so that other investigators can apply the clustering approach without having to match sensory stimulation conditions. However, it would be useful, perhaps as a supplementary figure, to know how well the classification would work if it also included the evoked activity metrics (SI_1 and SI_2). Airpuff stimuli are easy to apply so if the improvement were meaningful, others might want to use this as well.

2. The two sentences beginning with "Within the population of PTs, PCs in the..." on lines 155-158 are at first glance confusing. I understand the meaning here, but it reads as though the first sentence states that dendritic morphology did not differ by type, and the second sentence states that it did in fact differ. The authors may want to clarify.

Reviewer #2 (Remarks to the Author):

Pyramidal tract (PT) cells of layer 5 send axons to diverse brain regions. This paper addressed the following questions; Are PT cells composed of distinct groups with different combination of output targets?; If so, are the groups different in sensory responses? The authors answered yes to these questions, and their findings will contribute to understanding how the neocortex generate the subcortical outputs.

major:

1. The dendritic distribution were separately analyzed in L5A and L5B, which would be different between L5A and L5B PTs. Therefore, L5A and L5B PTs should be compared. Furthermore, in individual projection groups of PTs, L5a and L5b subgroups would be different in morphology and firing properties.

2. Fig 3b, line 148-154, For the PCA, what kind of morphological parameters were used? This Figure just says that L5 PTs are thick-tufted cells. Define the differences of slender- and thick-tufted PTs clearly. Are no differences among the 4 groups?

3. Were recorded cells labeled with a single tracer?

4. The authors have divided the PTs into the different projection groups. In other areas, PTs send axon collaterals to several subcortical regions simultaneously (e.g., . Levesque et al., Cereb Cortex, 1996 6:759; Kita, J Neurosci, 2012 32:5990). Could you discuss this point amore in Discussion?

minor:

1. Fig. 1g: How were double-labeled cells counted?

2. Fig, 1b, The depth distribution of L5 IT cells is shown to the right side. How could it be obtained?

3. Fig 2b, This panel is not necessary here. Important things related to Fig. 1b were not mentioned in the text.

4. line 181-186 Could you explain using the Table?

Reviewer #3 (Remarks to the Author):

In the manuscript by Rojas-Piloni & Guest et al., L5 pyramidal tract neurons in somatosensory cortex of rat are functionally and structurally characterized *in vivo* relative to four different subcortical target regions (posteriomedial nucleus of the thalamus, pontine nucleus, trigeminal nucleus of the brainstem, and superior colliculus). The data and analysis presented provides a useful insight into the function of specific classes of L5 pyramidal neurons. These data can help the field better understand the *in vivo* functional responses and related connectivity of L5 neurons. Overall, the manuscript makes a fair case for the relationship of connectivity, ongoing firing properties, morphology, and soma depth in cortex. Nevertheless, using these results as a means to predict the target of a given neuron in subsequent studies seems laborious and still leaves some doubt.

Comments on the manuscript:

- 1) A significant thrust of the manuscript is aimed at clustering PTs' physiology and morphology relative to a single target region. However, many neurons have multiple subcortical targets. What were the characteristics of double/triple labeled neurons? Did they have different characteristics, or did they fall into a target-specific cluster?
- 2) How can it be known that the analyzed neurons were only single-region projecting neurons? The injections would certainly not include the entire volume of the subcortical target areas. The overlap of different tracers could therefore be largely underestimated. Do you have a sense of what fraction of the different injected areas were filled by the retrograde tracers?
- 3) Were the same tracers used for a given region, or where they randomly injected into any given region? Is it known if the retrograde tracers can have an effect on firing properties of neurons? Do they all have a similar uptake efficiency?
- 4) How might anesthesia affect the firing patterns of the neurons tested during stimulation?

Minor issues:

- 1) It seems that the functional implications of these results are somewhat lacking in the discussion. Do you think the significant differences in dendritic path length in specific layers for each type of PT could correlate to what type of input these neurons prefer? The authors find differences among the PT stimulus response depending on target location. What could this mean in terms of processing in the wider circuitry?
- 2) In figure 2b, the upper panels could benefit from a label depicting which type of PT the recordings are from. It seems they correspond to the neurons labeled on the bottom panels, but having labels above would be easier to follow.
- 3) Figure 3c, how many neurons are averaged into the data shown?
- 4) Figure 3e, are the black circles with blue outline putative POM-projecting PTs simply based on their firing frequency? This seems rather speculative, even if they are considered "putative". If they could not be definitively identified, they should be left as such.
- 5) Line 703: "Intuitively, we calculated the mean and variability...." I would recommend removing the word 'intuitively'.

6) The word 'disjoint' is used often to mean separate populations of neurons. In the context of this work, this word seems odd. Perhaps, 'non-overlapping', 'different', 'separate' etc. could be used as well or in place of disjoint.

Reviewers' comments:

Reviewer #1 (Remarks to the Author):

In this manuscript, Rojas-Piloni, Guest et al. investigate how the structure and activity of pyramidal tract (PT) neurons in layer 5 (L5) of rat barrel cortex relate to their long-range projection targets. This is an important problem within the larger context of understanding how brain structure and circuitry relate to function. The authors injected retrograde tracers---up to 3 in individual animals----into three major subcortical target regions. In addition, they performed in vivo cell-attached electrophysiological recordings and filled recorded neurons with biocytin to reconstruct their soma location and dendritic morphologies with respect to the cortical layers, and to relate associated features to the long-range projection targets of each neuron. The authors find that PT neurons that project to the superior colliculus (SC-Ps), to the Pons (Pons-Ps), posterior-medial thalamus (Pom-Ps), and spinal trigeminal nucleus caudalis (SP5C-Ps) are largely overlapping in most individual properties, but contain distinct groups when their properties are considered together (i.e. when clustered in multidimensional space). First, the four types of PTs differ in the distributions of their soma locations within L5. For instance, SC-Ps tend to have a more shallow soma location, although the distributions were highly overlapping. The dendritic morphology also differed among classes. The authors first define morphology in terms of a first principal component (PC1) from principal component analysis (PCA) on a set of 21 metrics related to dendrite structure. This PCA distinguished the PTs of the present study from thin-tufted intratelencephalic (IT) L5 neurons, but did not distinguish among the subclasses of PT. However, a single measure---the distribution of dendritic length across cortical depth----did distinguish. (Dendritic length and the number of branch points were highly correlated, and offer a measure of dendrite complexity.) The spontaneous spike rates differed by PT type, with Pom-Ps showing higher levels of activity. Airpuff stimulus-evoked activity also differed among types, with all types showing similar stimulus onset responses, but different patterns of responses to the sustained and offset portions of the stimulus. The authors show that no one individual feature of the morphology or activity allows classification of the PTs by projection target, but that clustering in three dimensions---based on soma depth, PC1, and spontaneous activity---allow PT types to be classified with an accuracy of about 80%. This means that other investigators can with reasonable confidence infer the projection target of recorded PTs after recording spontaneous activity and reconstructing local (within column) morphology, without having to perform difficult retrograde labeling procedures such as those employed here. The authors offer Python-based analysis code to explicitly document, and to help readers apply, the classification method. I did not evaluate the code and associated files in detail, but the code looks clear and well documented, and its inclusion as part of this manuscript is a notable strength. The manuscript addresses an important problem. The quality of the data and analyses appear high. The manuscript is clearly written. I am happy to report that I have no major concerns, and that I think this will make a strong contribution.

We thank the reviewer for the positive evaluation of our manuscript. We have addressed the two suggestions raised and provide a detailed description of the resultant changes in line below. All changes throughout the revised manuscript are highlighted in red.

I have only two minor suggestion:

1. Clustering based on properties that do not require delivery of stimuli---i.e. spontaneous spike rate, depth, and morphology---makes sense, so that other investigators can apply the clustering approach without having to match sensory stimulation conditions. However, it would be useful, perhaps as a supplementary figure, to know how well the classification would work if it also included the evoked activity metrics (SI_1 and SI_2). Airpuff stimuli are easy to apply so if the improvement were meaningful, others might want to use this as well.

We have augmented the target prediction tool with a fourth option that allows uploading the PSTH of a recorded neuron. The tool will automatically compute the two similarity indices with

respect to our data and incorporate these results into the target prediction. In the revised version of the Supplementary Methods, we however emphasize that deviations from our experimental conditions may result in false assignment of the PTs' subcortical targets. We applied this new option to our data, which increased the prediction accuracy to 95%, compared to 86% when the evoked PSTHs were not included. We have added a sentence to the result section about this option to improve the target prediction (line: 274-276).

2. The two sentences beginning with “Within the population of PTs, PCs in the...” on lines 155-158 are at first glance confusing. I understand the meaning here, but it reads as though the first sentence states that dendritic morphology did not differ by type, and the second sentence states that it did in fact differ. The authors may want to clarify. We have now clarified the differences between a PT's morphology and its layer-specific dendrite distributions (lines: 160-170).

Reviewer #2 (Remarks to the Author):

Pyramidal tract (PT) cells of layer 5 send axons to diverse brain regions. This paper addressed the following questions; Are PT cells composed of distinct groups with different combination of output targets? If so, are the groups different in sensory responses? The authors answered yes to these questions, and their findings will contribute to understanding how the neocortex generate the subcortical outputs.

We thank the reviewer for the positive evaluation of our manuscript. We have addressed the issues raised and provide a detailed description of the resultant changes in line below. All changes throughout the revised manuscript are highlighted in red.

Major:

1. The dendritic distribution were separately analyzed in L5A and L5B, which would be different between L5A and L5B PTs. Therefore, L5A and L5B PTs should be compared. Furthermore, in individual projection groups of PTs, L5a and L5b subgroups would be different in morphology and firing properties.

*We agree with the reviewer that it could be possible that neurons within each L5 cell type (i.e. slender-tufted ITs and thick-tufted PTs) may have different structural and/or functional properties, depending on whether their somata are located within L5A or L5B, respectively. In the revised version of the manuscript, we have therefore analyzed all target-related properties of PTs as a function of soma depth. We have added a new supplementary figure (**Fig. S2**) which shows the relationships between each PT's registered soma depth and its (1) dendrite morphology, (2) dendrite distribution, (3) ongoing spike rate and (4) PSTH similarity. None of these four target-related properties correlates with soma depth. Consequently, grouping PTs in L5A vs. L5B revealed no significant differences in terms of structure or in vivo function. In the revised manuscript, we have augmented the result section with the following quantifications: First (line: 148-153), we compared the soma depth distribution of reconstructed PTs with the one obtained by quantification of the retrograde injections (i.e. **Fig. 1b**). The median soma depth across the sample of reconstructed neurons is close to the peak of the vertical profile of the retrogradely labelled PTs. The vertical extents of the two differently obtained PT soma distributions are also very similar. Thus, the present sample of recorded/reconstructed PTs can be regarded as representative for the distribution of PTs across the entire depth of L5. Second (line: 160-165), the soma depth distributions were not significantly different between reconstructed PTs when grouping them by their targets. Furthermore, we added the median soma depth for each group to the average dendrite distribution in **Fig. 3c**. Thus, the soma depth distributions between the four PT groups are not significantly different, but the respective dendrite distributions that these populations give rise to, differ significantly in a layer-specific manner.*

2. Fig 3b, line 148-154, For the PCA, what kind of morphological parameters were used?

This Figure just says that L5 PTs are thick-tufted cells. Define the differences of slender- and thick-tufted PTs clearly. Are no differences among the 4 groups?

We added a new supplementary table (Table S1), which shows all parameters, the respective means and SDs for each of the four groups, as well as the P-values from 1-way ANOVA tests, in which we compared all parameters between the four groups. Further, in the legend of Table S2, we provide a brief definition of each morphological parameter. The only parameter that was at trend level was the number of branch points, which originated from the group of POM-projectors that showed in general less branch points compared to PTs from the other 3 groups. This is stated in the result section and is discussed, for example with respect to the more specific intrinsic properties of POM-Ps. A full list of all 21 parameters for slender- and thick-tufted neurons can be obtained from Table 1 in (Narayanan et al., 2015). The new supplementary table is ordered identical to Table 1 of the Narayanan et al., study, which allows to directly compare the 21 parameters of the present dataset (and for each PT group) with those previously reported for slender- and thick-tufted cells. Table S1 is referenced both in the result section (lines: 155 and 163) and in the legend of Figure 3 (line: 424).

3. Were recorded cells labeled with a single tracer?

Three of the 23 in vivo recorded PTs were labeled with two retrograde markers: one projected to the POM and Pons, two projected to the POM and SC. Our cluster analysis had revealed that these three neurons show structural and functional properties that were related consistently to the properties of only one of the respective projection groups. In the previous version of the manuscript we had thus grouped these dual-Ps with the respective PTs as assigned by the clustering. In the revised version of the manuscript, we have now excluded the three dual-Ps from all target-specific analyses and analyzed their properties separately. We recalculated all numbers and highlighted the new values in red throughout the result section and in Table 1. Further, we updated figures 3-5, excluding the dual-Ps from all panels that show average properties of a projection group. In panels that show the individual data points (also in the new supplementary Fig. S2), dual-Ps have now two colors, with the center color representing the target with which they share their properties. Furthermore, we added two new paragraphs to the result section, where we report the structure, function and classification of the dual-Ps (lines: 226-235) and their stimulus-evoked responses (lines 273-276), respectively. We also discuss projections to multiple targets in more detail now (see below).

4. The authors have divided the PTs into the different projection groups. In other areas, PTs send axon collaterals to several subcortical regions simultaneously (e.g., . Levesque et al., Cereb Cortex, 1996 6:759; Kita, J Neurosci, 2012 32:5990). Could you discuss this point amore in Discussion?

In the previous version of the manuscript, we had stated that our observations do not imply that PTs in vS1 project only to a single subcortical target area. In the revised version, we have now stressed this point further. First, we rephrased the result section (lines 114-115) because the way how we reported the fraction of dual-Ps may have been ambiguous. We now state that on average 19% of the PTs within each group project additionally to each of the other three targets. I.e. the probability that e.g. a POM-P also innervates the SC or Pons or Sp5C is 0.19 ± 0.12 . Our estimates are within range of a recent study (0.32 ± 0.09) that reconstructed axons of individual thy1-positive neurons in L5 of mouse vS1 (Guo et al., 2017). The mouse study did however not differentiate between different subnuclei within the four main target regions of vS1, which is likely to explain the slightly higher overlap fractions. For example, in line with a previous report (Hattox & Nelson 20007), we find that there is almost no overlap between POM- and Sp5C-Ps (<2%). In contrast, the fraction of corticothalamic PTs that also project to the spinal trigeminal tract was reported in the mouse study as ~23%. However, PTs in vS1 project to different subnuclei of the spinal trigeminal tract. PTs that for example project to the Pr5 or Sp5lc (see Smith et al., 2015) may thus have more frequent projections to the POM than those projecting to the Sp5C. We have now added the single mouse study to the references and used it to provide a more detailed discussion of the various targets that PTs can innervate in addition to those investigated here (lines: 282-295). Specifically, in agreement

with our quantifications of retrogradely labeled PTs, the single axon tracing study shows that the majority of PTs in vS1 project to only a single subcortical target, and that the vast majority of PTs (>80%) projects to one of the four main brain regions that comprise the targets investigated here. We thus conclude that more than 80% of the PTs in vS1 project to the four targets investigated here, that PTs are largely non-overlapping with respect to these four targets, but that they could project to one (or more) additional targets (i.e. different subnuclei within the four main target regions and/or different brain regions).

We also emphasize that PTs can overlap with respect to the four targets investigated here, which is quantified in **Table S2** and stressed in the revised manuscript as follows: we modified **Fig. 1g** (lines: 405-409) to show the number of PTs projecting to two of the injected targets and we now analyzed the structure and function of PTs with two targets separately from those with one target (see above).

minor:

1. Fig. 1g: How were double-labeled cells counted?

We updated the method section as requested (lines: 702-706).

2. Fig, 1b, The depth distribution of L5 IT cells is shown to the right side. How could it be obtained?

The distribution of ITs was determined from previous studies in which we determined the number and distribution of L5 slender-tufted neurons. We added this information to the figure legend (lines: 399-401 and 408-409).

3. Fig 2b, This panel is not necessary here. Important things related to Fig. 1b were not mentioned in the text.

The figure is now referenced in the text (line 136) where we state that we ‘...measured ongoing and sensory-evoked spiking in anesthetized young adult rats...’. We believe that it is important to show a few examples with ‘raw’ spiking data vs. morphology and target.

4. line 181-186 Could you explain using the Table?

We added a reference to Table 1 at this location in the revised manuscript (line 199).

Reviewer #3 (Remarks to the Author):

In the manuscript by Rojas-Piloni & Guest et al., L5 pyramidal tract neurons in somatosensory cortex of rat are functionally and structurally characterized in vivo relative to four different subcortical target regions (posteriomedial nucleus of the thalamus, pontine nucleus, trigeminal nucleus of the brainstem, and superior colliculus). The data and analysis presented provides a useful insight into the function of specific classes of L5 pyramidal neurons. These data can help the field better understand the in vivo functional responses and related connectivity of L5 neurons. Overall, the manuscript makes a fair case for the relationship of connectivity, ongoing firing properties, morphology, and soma depth in cortex. Nevertheless, using these results as a means to predict the target of a given neuron in subsequent studies seems laborious and still leaves some doubt.

We thank the reviewer for the positive evaluation of our manuscript. We have addressed the issues raised and provide a detailed description of the resultant changes in line below. All changes throughout the revised manuscript are highlighted in red.

Comments on the manuscript:

1) A significant thrust of the manuscript is aimed at clustering PTs’ physiology and morphology relative to a single target region. However, many neurons have multiple subcortical targets. What were the characteristics of double/triple labeled neurons? Did they have different characteristics, or did they fall into a target-specific cluster?

In the previous version of the manuscript, we had stated that our observations do not imply that PTs in vS1 project only to a single subcortical target area. In the revised version, we have now stressed this point further. First, we rephrased the result section (lines 114-115) because the way how we reported the fraction of dual-Ps may have been ambiguous. We now state that on average 19% of the PTs within each group project additionally to each of the other three targets. I.e. the probability that e.g. a POM-P also innervates the SC or Pons or Sp5C is 0.19 ± 0.12 . Our estimates are within range of a recent study (0.32 ± 0.09) that reconstructed axons of individual thy1-positive neurons in L5 of mouse vS1 (Guo et al., 2017). The mouse study did however not differentiate between different subnuclei within the four main target regions of vS1, which is likely to explain the slightly higher overlap fractions. For example, in line with a previous report (Hattox & Nelson 2007), we find that there is almost no overlap between POM- and Sp5C-Ps (<2%). In contrast, the fraction of corticothalamic PTs that also project to the spinal trigeminal tract was reported in the mouse study as ~23%. However, PTs in vS1 project to different subnuclei of the spinal trigeminal tract. PTs that for example project to the Pr5 or Sp5lc (see Smith et al., 2015) may thus have more frequent projections to the POM than those projecting to the Sp5C. We have now added the single mouse study to the references and used it to provide a more detailed discussion of the various targets that PTs can innervate in addition to those investigated here (lines: 282-295). Specifically, in agreement with our quantifications of retrogradely labeled PTs, the single axon tracing study shows that the majority of PTs in vS1 project to only a single subcortical target, and that the vast majority of PTs (>80%) projects to one of the four main brain regions that comprise the targets investigated here. We thus conclude that more than 80% of the PTs in vS1 project to the four targets investigated here, that PTs are largely non-overlapping with respect to these four targets, but that they could project to one (or more) additional targets (i.e. different subnuclei within the four main target regions and/or different brain regions).

*We also emphasize now that PTs can overlap with respect to the four targets investigated here, which is quantified in **Table S2** and stressed in the revised manuscript as follows: we modified **Fig. 1g** (lines: 405-409) to show the number of PTs projecting to two of the injected targets and we now analyzed the structure and function of PTs with two targets separately from those with one target. Specifically, three of the 23 in vivo recorded PTs were labeled with two retrograde markers: one projected to the POM and Pons, two projected to the POM and SC. Our cluster analysis had revealed that these three neurons show structural and functional properties that were related consistently to the properties of only one of the respective projection groups. In the previous version of the manuscript we had thus grouped these dual-Ps with the respective PTs as assigned by the clustering. In the revised version of the manuscript, we have now excluded the three dual-Ps from all target-specific analyses and analyzed their properties separately. We recalculated all numbers and highlighted the new values in red throughout the result section and in **Table 1**. Further, we updated **figures 3-5**, excluding the dual-Ps from all panels that show average properties of a projection group. In panels that show the individual data points (also in the new supplementary **Fig. S2**), dual-Ps have now two colors, with the center color representing the target with which they share their properties. Furthermore, we added two new paragraphs to the result section, where we report the structure, function and classification of the dual-Ps (lines: 226-235) and their stimulus-evoked responses (lines 273-276), respectively.*

2) How can it be known that the analyzed neurons were only single-region projecting neurons? The injections would certainly not include the entire volume of the subcortical target areas. The overlap of different tracers could therefore be largely underestimated. Do you have a sense of what fraction of the different injected areas were filled by the retrograde tracers?

We agree with the reviewer that the interpretation of our data critically depends on the quality of the injections. In the revised manuscript, we have thus added a quantification of the size and location of the individual injection sites with respect to the target areas. The quantification confirmed that injections were precise and large enough to cover on average 90% of the

targets. We state these numbers now in the result section, including the min-max ranges of the distances to the target centers and overlaps between injection sites and targets (lines 93-96). We also extended the method section and now describe how we quantified the injection sites (lines 613-622), and explain in more detail our experimental procedures to obtain precise and reliable injections (i.e. electronic leveling of the animal's head with respect to the medial-lateral and anterior-posterior planes; lines 593-602). Further, we added a new panel to **Fig. S1**, which illustrates the average injection sites that correspond to the average soma distributions of PTs in **Fig. 1b**. Because the injections are reliably centered on the targets and cover ~90% of the target areas, and because the resultant probabilities of dual-Ps are similar to those obtained from single axon tracings in mouse vS1, we argue that our results represent realistic estimates of the fraction of dual-Ps (i.e. with respect to the targets investigated here).

3) Were the same tracers used for a given region, or where they randomly injected into any given region?

We used the three tracers equivalently, and randomly selected combinations of retrograde tracers for triple injections. We state this now in the method section and provide the number of experiments for each target/tracer combination (lines: 606-611).

Do they all have a similar uptake efficiency?

The reason why we were able to use the three retrograde tracers equivalently was shown in **Fig. 1c**. We had injected FG and CTB via the same pipette into the Sp5C. In these control experiments, virtually every cell in vS1 was labeled by both markers. We now emphasize this control experiment further in the revised manuscript (lines 105-107; 401-402).

Is it known if the retrograde tracers can have an effect on firing properties of neurons?

To the best of our knowledge, the retrograde tracers have no effect on firing properties. The tracers had been used previously to study relationships between intrinsic physiological properties and long-range targets *in vitro*, or to study relationships between sensory-evoked calcium-transients and long-range targets *in vivo*. Moreover, the ongoing spike rates and whisker-evoked PSTHs reported here, are qualitatively and quantitatively very similar to those reported previously for L5 thick-tufted neurons (e.g. as shown in **Table 1**). Moreover, 20 of our 43 morphologically identified PTs were not labeled by retrograde markers, and can thus be regarded as a control group (black markers in **Fig. 3** and **5**). We state throughout the result section that ongoing spike rates or whisker-evoked responses were not different between retrogradely labeled PTs and those without retrograde labeling; e.g. lines 187-189.

4) How might anesthesia affect the firing patterns of the neurons tested during stimulation?

It is to be expected that changes in experimental conditions, e.g. anesthetized vs. awake, or passive vs. active touch stimuli, will impact the sensory-evoked responses of PTs. For that reason, we had excluded the evoked data from the cluster analysis and the resultant target-prediction tool. We argued that the three parameters of the cluster tool are expected to be largely independent of experimental conditions, because ongoing spike rates were shown to be quantitatively similar for different experimental conditions. For example, ongoing spike rates of PTs in deeply anesthetized animals (e.g. the present study, but see also de Kock et al., 2006) is very similar to those of PTs in fentanyl-sedated (e.g. Constantinople & Bruno, 2013) or awake rats (de Kock & Sakmann, 2009). The degree to which whisker-evoked spike patterns (i.e. as evoked by the same stimulus) change across PTs for different experimental conditions, and whether the changes affect PTs with different targets in similar ways, remains to be investigated. We therefore suggest that the clustering method will help to facilitate such investigations in the future. What we can say is that PTs in deeply anesthetized, sedated and awake animals show qualitatively similar responses to passive (and active) whisker touches. Specifically, PTs show a reliable increase of spike rates at short-latencies and often sustained spiking activity during the stimulation period.

Please note that even though evoked spiking patterns will be affected by experimental conditions, we followed the suggestion by reviewer #1 and included the PSTH similarity indices as optional into the clustering tool. For the present experimental conditions, including evoked spiking improved the predictability of the targets to 95% confidence (lines 274-276). However, in the respective section of the supplementary methods, we stress that experimental conditions will affect evoked responses and that this clustering option should hence be use cautiously (i.e. ideally only when our experimental conditions are met).

Minor issues:

1) It seems that the functional implications of these results are somewhat lacking in the discussion. Do you think the significant differences in dendritic path length in specific layers for each type of PT could correlate to what type of input these neurons prefer? The authors find differences among the PT stimulus response depending on target location. What could this mean in terms of processing in the wider circuitry?

We discuss that target-related laminar soma-dendrite distributions are likely to reflect differences in synaptic input (lines 334-341). These differences could be twofold. First, all PTs are synaptically innervated by the same populations but at different locations along the dendrites. The location-specific distributions of ion-channels along PT dendrites could hence result in target-related responses, even though all PTs receive the same input. Second, PTs are synaptically innervated (at least to some degree) by different populations, for example depending on the degree to which their respective soma-dendrite distributions overlap with layer-specific projections patterns of long-range input populations. These two possibilities are not mutually exclusive and could indicate that PTs extract information from a given sensory stimulus that is most relevant for its respective long-range targets. In the revised version, we have now added a discussion about functions that these circuits may be involved in (e.g. whisker motor control; lines: 352-360).

2) In figure 2b, the upper panels could benefit from a label depicting which type of PT the recordings are from. It seems they correspond to the neurons labeled on the bottom panels, but having labels above would be easier to follow.

We changed this as suggested.

3) Figure 3c, how many neurons are averaged into the data shown?

This is now stated in the figure legend (line 421-422). All sample sizes are also provided in Tables 1 and S1.

4) Figure 3e, are the black circles with blue outline putative P0m-projecting PTs simply based on their firing frequency? This seems rather speculative, even if they are considered “putative”. If they could not be definitively identified, they should be left as such.

We agree with the reviewer and removed the blue outlines.

5) Line 703: “Intuitively, we calculated the mean and variability....” I would recommend removing the word ‘intuitively’.

We changed this as suggested.

6) The word ‘disjoint’ is used often to mean separate populations of neurons. In the context of this work, this word seems odd. Perhaps, ‘non-overlapping’, ‘different’, ‘separate’ etc. could be used as well or in place of disjoint.

We changed this as suggested.

REVIEWERS' COMMENTS:

Reviewer #1 (Remarks to the Author):

I continue to think this is a strong manuscript, now even more so after revision. The authors have addressed my previous minor suggestions.

There is a typo on line 274: "Combing" should be "Combining".

Reviewer #2 (Remarks to the Author):

Dear Authors:

By additional analysis and discussion, following important facts have been successfully added to the paper:

1. Morphological properties and sensory responses of PT cells depends on their axonal targets, but not on the soma position in layer 5.
2. Most PT cells in S1 project one of four target regions: thalamus, midbrain, pons and spinal trigeminal tract.

This is a very important paper because it has clearly revealed that PT cells are composed of functionally independent groups projecting to different extra-telencephalic targets.

Reviewer #3 (Remarks to the Author):

The authors have put in substantial effort to address the comments to a satisfactory degree. I am satisfied with the changes made.

That said, one minor addition that might improve clarity in Fig 4D would be to include that the Dual-Ps are the outlined boxes. Perhaps:

"d. Classification of PTs using all three target-related properties (dual-Ps are outlined)."

Reviewers' comments:**Reviewer #1 (Remarks to the Author):**

I continue to think this is a strong manuscript, now even more so after revision. The authors have addressed my previous minor suggestions.

We thank the reviewer for the continuous support of our manuscript.

There is a typo on line 274: "Combing" should be "Combining".

We have corrected all typos throughout the manuscript.

Reviewer #2 (Remarks to the Author):

By additional analysis and discussion, following important facts have been successfully added to the paper:

1. Morphological properties and sensory responses of PT cells depends on their axonal targets, but not on the soma position in layer 5.
2. Most PT cells in S1 project one of four target regions: thalamus, midbrain, pons and spinal trigeminal tract.

This is a very important paper because it has clearly revealed that PT cells are composed of functionally independent groups projecting to different extra-telencephalic targets.

We thank the reviewer for the positive evaluation and continuous support of our manuscript.

Reviewer #3 (Remarks to the Author):

The authors have put in substantial effort to address the comments to a satisfactory degree. I am satisfied with the changes made.

We thank the reviewer for the positive evaluation and support of our manuscript.

That said, one minor addition that might improve clarity in Fig 4D would be to include that the Dual-Ps are the outlined boxes. Perhaps: "d. Classification of PTs using all three target-related properties (dual-Ps are outlined)."

We changed the figure legend as suggested.